# Pregnant Women with Multiple Sclerosis: An Overview of Gene Expression and Molecular Interaction Using Bioinformatics Analysis

**DOI:** 10.3390/ijms25126741

**Published:** 2024-06-19

**Authors:** Jazmin Marquez-Pedroza, Martha Rocio Hernández-Preciado, Edgar Ricardo Valdivia-Tangarife, Francisco J. Alvarez-Padilla, Mario Alberto Mireles-Ramírez, Blanca Miriam Torres-Mendoza

**Affiliations:** 1Neurosciences Division, Western Biomedical Research Center, Mexican Institute of Social Security, Guadalajara 44340, Mexico; jaz180688@gmail.com; 2Department of Philosophical and Methodological Disciplines, University Health Sciences Center, University of Guadalajara, Guadalajara 44340, Mexico; mrociohp@hotmail.com; 3Department of Neurosciences, University Health Sciences Center, University of Guadalajara, Guadalajara 44340, Mexico; ricardovaldiviatangarife@outlook.com; 4Translational Bioengineering Department, University Center of Exact Sciences and Engineering, University of Guadalajara, Guadalajara 44430, Mexico; francisco.alvarez@academicos.udg.mx; 5High Specialty Medical Unit, Western National Medical Center, Mexican Institute of Social Security, Guadalajara 44340, Mexico; drmmireles@gmail.com

**Keywords:** multiple sclerosis, pregnancy, gene expression, molecular interaction, bioinformatics

## Abstract

Multiple sclerosis (MS) is a common disease in young women of reproductive age, characterized by demyelination of the central nervous system (CNS). Understanding how genes related to MS are expressed during pregnancy can provide insights into the potential mechanisms by which pregnancy affects the course of this disease. This review article presents evidence-based studies on these patients’ gene expression patterns. In addition, it constructs interaction networks using bioinformatics tools, such as STRING and KEGG pathways, to understand the molecular role of each of these genes. Bioinformatics research identified 25 genes and 21 signaling pathways, which allows us to understand pregnancy patients’ genetic and biological phenomena and formulate new questions about MS during pregnancy.

## 1. Introduction

MS is an autoimmune-mediated neurodegenerative disease of the CNS characterized by demyelinating lesions and axonal injury [1]. It is a complex disease whose exact etiology remains uncertain [2]. Therapeutic options are disease-modifying therapies (DMTs), which have the highest efficacy in treating active MS [3]. Establishing possible genetic triggers of disease could make way for pathway-based medicine, immunotherapies, and other treatments that target specific molecules [4,5].

Women of reproductive age are typically affected, with the median age of disease onset being around 30 years [3]. The impact on functionality or disability in patients with MS during pregnancy is controversial and variable. Some studies suggest that during pregnancy, many women experience a decrease in disease activity (pregnancy protective effect) and may have a temporary improvement in their neurological symptoms [6]. On the other hand, some women with MS may experience a fall or worsening of their symptoms after childbirth (postpartum effect) [7].

The Pregnancy in MS Study (PRIMS) showed that the relapse rate dropped to 70% during the third trimester and increased in the same way in the first three months postpartum. Risk factors for a postpartum relapse are a high rate of relapse in the year before pregnancy, high disability before pregnancy, and suffering a relapse during pregnancy [6].

Other studies have shown a significant decrease in relapses during pregnancy, though the mechanisms behind this remain undisclosed [6,7]. There is no evidence to support the claim that pregnancy itself permanently changes or modifies a woman’s ability to function with MS (Table 1).

MS patients with less active disease or disability before pregnancy are likely to have a decrease on their Expanded Disability Status Scale (EDSS) during pregnancy than more impaired MS patients or those with more active disease [14,15]. Changes in the immune system during pregnancy could result in a decrease in MS activity during pregnancy and a brief improvement in EDSS [11].

## 2. Treatment during Pregnancy

The selection of MS treatments during pregnancy can be influenced by various factors, including the progression of the disease, breastfeeding, methods of administration, mechanisms of action, and their effectiveness and safety profiles. Certain medications commonly used to manage MS, such as teriflunomide, fingolimod, siponimod, and cladribine, may not be considered safe during pregnancy [16].

Some medications have been proven safe to use, but only during certain stages of pregnancy (glatiramer acetate and interferon-betas during the first trimester), and other therapies require more studies to determine the effects during pregnancy [17]. It is recommended to discontinue DMTs during pregnancy; however, this produces a greater risk of relapse [11].

## 3. Expression of Genes Related to MS in Pregnancy

Like other autoimmune diseases, the molecular mechanisms described include the altered expression and regulation of multiple genes, such as chemokine receptors and interleukins [1]. MS is considered a Th1 cell-mediated disease. The activation of T cells and Th1 and Th2 cells express distinct patterns of chemokine receptors that play a role in the pathogenesis of inflammation. During the second and third trimesters of pregnancy, there is an increase of Th2 in peripheral blood leukocytes relative to Th1 cells. Hormone-induced changes in T cell chemokine receptors are mainly responsible for the leukocyte recruitment mechanisms, prioritizing determined leukocyte subtypes needed at the maternal–fetal interface [18].

Some chemokine receptors expressed in T cells, such as CXCR3 and CCR5, in MS play a crucial role in demyelination by promoting T cell accumulation in the CNS [18]. The evidence found decreased the receptor CXCR3 expression of T CD4 and T CD8 cells, reaching statistical significance during the third trimester. Likewise, CXCR4 expression by T CD4 cells presented a statistically significant increase in the third trimester [19].

Additionally, the expression of IL-10 mRNA by peripheral blood mononuclear cells (PBMC) increased during pregnancy and decreased during postpartum. Contrarily, IFN-γ levels by PBMC decreased during pregnancy and increased postpartum. Such ratios showed statistically significant differences. It would be worth further studying a larger population to validate these differences [18].

Numerous other inflammation-related genes, such as *SOCS2*, *TNFAIP3*, *NR4A2*, *CXCR4*, *POLR2J*, *FAM49B*, and *STAG3L1*, have shown altered expression during pregnancy in patients with MS. Interestingly, research has found that genes *CXCR4*, *TNFAIP3*, *SOCS2*, and *NR4A2* show elevated levels in patients with MS during early pregnancy. In the same trimester of pregnancy, healthy controls downregulated these same transcripts. The primary explanation for this is the activity of the *SOCS2* gene, which inhibits prolactin transduction and downregulates growth hormone. As a result, *SOCS2* inhibition is essential for the development of pregnancy. On the other hand, *TNFAIP3* is necessary for trophoblast invasion; although its upregulation is crucial during the early stages of pregnancy, the results are dangerous in the third trimester. Furthermore, CXCR4 is expressed in early pregnancy as it plays essential roles in embryogenesis and organogenesis [20].

The CD64 Fc-receptor (*FcgR1a*) undergoes a relevant change during pregnancy in MS patients. A 2010 study revealed an increased expression of CD64 in pregnant patients with MS during the third semester. However, because such upregulation is also present in healthy pregnant patients, it was concluded that it cannot be suggested that the upregulation is due to MS [21].

NK cells are potent immunoregulatory cells that could play a protective role in autoimmunity. A higher expression of HLA-E in T cells has an inhibitory effect on NK cells. HLA-E is upregulated in MS patients in white matter lesions, endothelial cells, and astrocytes [22].

CD4 T cells balance immunity and tolerance. A 2022 study elucidated their importance in autoimmune diseases and the changes in their regulation during pregnancy. Pregnancy is associated with hypermethylation more than hypomethylation; thus, a more significant “inhibition” of T cell-mediated immunity. Some of the hypermethylated regulators in pregnancy are CD28, critical in T cell activation, and CD86, co-stimulatory of T cells when expressed in memory effector T cells [23].

Pregnancy hormones, mainly estrogen, have been considered the main drivers of the modulation of the immune system during pregnancy. The methylation of essential regulators, such as CD4 T cells, CD8 T cells, and inflammatory mediators, peaks in the third trimester and reverses in the postpartum period. The rebound pattern following delivery is consistent with this declaration. However, it represented normal immune system adaptation, with only slight differences noted between the MS patients and controls [23] (Table 2).

Further studies to evaluate methylation or epigenetic modifications of specific CNS immune modulators would be of great value.

## 4. Materials and Methods

### Bioinformatics Data Tools: Protein–Protein Interactions among Associated Genes

This review identified 25 genes related to various aspects of MS. Consequently, we were eager to understand the molecular role of each of these genes. To achieve this, we constructed an interaction network using the bioinformatics tool STRING database version 12.0 (available online: https://string-db.org/ accessed on 8 September 2023) [26] and leveraged KEGG pathway databases (available online: https://www.genome.jp/kegg/pathway.html accessed on 8 September 2023) [27]. To ensure reliable protein interactions, we set a minimum confidence score of 0.9 and considered interactions significant if they had an FDR of less than 0.25.

For the creation of the Sankey Plot, a novel visualization tool, which is depicted in Figure 1, we utilized the ggplot2 package version 3.5.1 (available online: https://cran.r-project.org/web/packages/ggplot2/index.html, accessed on 8 September 2023) [28], as well as the alluvial package (available online: https://cran.r-project.org/web/packages/alluvial/vignettes/alluvial.html, accessed on 10 September 2023) [29]. We conducted this analysis using R version 4.4.1 (available online: https://cran.r-project.org/bin/windows/base/old/, accessed on 8 September 2023) [30].

## 5. Results

### Molecular Interaction of Genes

After calculating the associations between the 25 genes related to MS and various signaling pathways, we considered interactions with a false discovery rate (FDR) ≤ 0.25 and a *p*-value ≤ 0.05 to construct the Sankey Plot shown in Figure 1. With these results, we provide essential information that contributes to understanding the molecular role of each of these genes in the context of the disease.

Our analysis has revealed that 25 genes interacted with 21 distinct signaling pathways. These interactions were diverse, and some were specific to a gene and a particular pathway. For instance, our results indicate interactions between herpes simplex virus 1 infection and *ZNF577*, zinc transport and *TMC8*, p53 signaling and *TP53*, complement system and *PTX3*, hematopoietic cell lineage and CD38, hemostasis and *FAM49B*, RNA polymerase and *POLR2J*, and cellular senescence and *ZFP36L1*.

We also discovered another group of pathways associated with two or more genes, including prolactin signaling with *JAK2*, *STAT1*, and SOCS2; Th1 and Th2 cell differentiation with *JAK2*, *STAT1*, *CXCL8*, and *INFG*; *TNF* signaling with *TNFAIP3*, *CXCL2*, and *CXCL8*; cell adhesion molecules with *PDCD1LG2* and CD274; parathyroid hormone synthesis, secretion, and action with *CXCL2*, *CXCL8*, and *CXCR4*; NOD-like receptor signaling with *PRKCB*, *STAT2*, *TNFAIP3*, *JAK2*, *CXCL2*, and *CXCL8*; NF-kappa B signaling with *TNFAIP3*, *STAT1*, *CXCL2*, and *CXCL8*; Th17 cell differentiation with *JAK2*, *STAT1*, and *INFG*; *PD-L1* expression and *PD-1* checkpoint pathways in cancer with CD274, *ANKRD22*, *VAV3*, *JAK2*, *STAT1*, and *INFG*; T cell receptors with *VAV3*, *CD28*, *INFG*, and *IL10*; T cell receptor signaling with *VAV3*, *CD28*, *INFG*, and *IL10*; FoxO signaling with *PLK1* and IL10; JAK-STAT signaling with *JAK2*, *STAT1*, *SOCS2*, *INFG*, and *IL10*; and cytokine–cytokine receptors with CXCL2, CXCL8, INFG, and IL10.

Notably, the pathways with the most significant interactions with our genes of interest were parathyroid hormone synthesis, NOD-like receptor signaling, NF-kappa B signaling, PD-1 checkpoint pathways in cancer, T cell receptor signaling, JAK-STAT signaling, and cytokine–cytokine receptors. These findings provide important insights into the molecular and cellular processes that underlie various biological phenomena and have significant implications for the understanding and treatment of MS (Figure 2 and Figure 3).

## 6. Discussion

We identified 25 genes and 21 pathways that may be involved in disease changes in pregnant women. The involvement of these mechanisms must be studied in greater depth since little information is available on their participation in MS. We show a general summary of the possible involvement of the central genes, proteins, and molecules obtained from this analysis. The regulation of inflammation supports successful reproduction and positively influences MS [20]. *TMC8* is a coding gene that could regulate CD4+ T cells. *ZNF577* is a gene encoding zinc finger proteins that participate in transcriptional regulation. Some studies have reported changes in the epigenetic patterns of CD4 T cells during pregnancy, which could affect the activity of MS [31]. *ZNF577* participants in the transcriptional regulation of the expression of HSV-1.

The role of HSV-1 in MS is controversial and the molecular interaction is unclear. The suggested relationship indicates that latent infection and the host’s inflammatory response to the CNS may exacerbate the disease. Children with the DRB1*15 allele and HSV-1 seropositivity are associated with an increased risk of MS compared to patients without the allele. In addition, there could be molecular mimicry, which consists of the cross-recognition of viral antigens and autoantigens by T cells. The T cell receptor (Hy.1B11) recognizes HSV-1 and the autoantigen myelin basic protein, which is normally expressed in neurons [32].

PTX3 is a soluble pattern recognition receptor crucial in modulating the inflammatory response and activating the complement system [33]. Longitudinal PTX3 levels during pregnancy are scarce. The first trimester of pregnancy is proinflammatory to facilitate blastocyst implantation. It has been suggested that elevated levels of PTX3 characterize the entire period of pregnancy. During the third trimester, the immune system is altered to a more proinflammatory state essential for birth, which could be reflected in the increase in PTX3 levels in late pregnancy [34]. In vitro studies show that CD38 acts as a regulator of neuroinflammatory processes. CD38 is expressed in B cell precursors, germinal center B cells, and plasma cells. Likewise, the expression of CD38 in NK cells has been linked to activation, cytotoxic activity induction, and IFN-γ secretion [35]. A study in pregnant rats demonstrated that the expression of CD38 is higher at parturition compared with days 14–17 of pregnancy. This increase in estrogen level is most likely involved in the regulation of CD38 expression and differential regulation of its enzyme activity [36]. The upregulation of *FAM49B* has been associated with the inhibition of T cell activation in patients with MS. *FAM49B* expression is deregulated before pregnancy and reversed during pregnancy [37]. In patients with MS, *POLR2J* upregulates inflammatory processes and plays a crucial role [20]. The gene expression was dysregulated before gestation and reverted by the pregnancy process [37]. *ZFP36L1* is a transcription factor involved in cell activation and has an anti-inflammatory role. *ZFP36L1* transcript levels are lower in MS patients compared with healthy controls [38]. *ZFP36L1* in mice is critical for normal fetoplacental development and fetal survival [39]. MS is mediated by both autoreactive Th1 and Th17 cells. The JAK-STAT pathway leads to Th1 and Th17 differentiation, which is involved in cell proliferation, differentiation, apoptosis, and immune regulation [40]. Th17 cells are associated with a higher expression of *JAK2*. Once Th17 cells complete their IL-23-dependent expansion, they become susceptible to the *STAT1*-dependent antiproliferative effect [41].

The JAK-STAT signaling is switched off by a particular SOCS [42]. *SOCS2* inhibits signal transduction induced by several cytokines, including IL-6. Also, *SOCS2* negatively regulates growth hormone action and inhibits prolactin signal transduction; thus, its downregulation is essential for a successful pregnancy [20].

Aberrant activation or phosphorylation of the JAK/STAT signaling pathway components has been implicated in increased transcription of the inflammation-associated genes and many neurodegenerative disorders [43]. CXCL8 is an inflammatory chemokine, and its related receptors are normally expressed at low levels in the CNS but increase their expression during neuroinflammation. CXCL2 signals CXCR2, activating STAT3, which contributes to neutrophil migration [44]. Additionally, CXCL8 regulates endothelial adhesion, chemotaxis, and activation of other leukocytes, monocytes, CD8+ T cells, and mast cells. CXCL8 can also directly bind to TNF-stimulated gene/protein-6, and this blinding inhibits neutrophil migration [45]. CXCL8 and CXCL2 regulate the function of *TNFAIP3T*. This protein protects against tumor necrosis factor (TNF)-mediated apoptosis and inhibits inflammation by blocking the transcription factor NF-κB [46].

The NOD-like receptor (NLR) is a family of proteins formed by fourteen members. Some NLRs are known to form multi-protein complexes known as inflammasomes. NLR facilitates the cleavage and activation of caspase-1, which mediates the cleavage of the proinflammatory cytokines IL-1β and IL-18 into their active and secreted forms [47]. CXCL1 and CXCL2 can activate NLRP3 inflammasome in macrophages through the CXCR2. Also, protein kinase C-B (PRKCB) is required for phosphorylation and inflammasome activation [48]. CXCL8 regulates the level of PTH secreted by the parathyroid. The parathyroid might be affected by a hormone whose production is stimulated during inflammation [49].

On the other hand, the PD-1/PD-L1 pathway is a negative regulator of immune responses, regulating the expansion, differentiation, and activation of immune cells. The PD-1/PDL-1 pathway inhibits STAT1 and produces a negative regulation of the cytokines IFN-γ, TNF-α, and IL-2. The inhibition of PD-1 on lymphocytes in the acute phase of MS significantly increases the proliferation of CD4 + T cells and CD8 + T cells. Therefore, the PD-1/PD-L1 pathway plays an immunosuppressive role in MS [50]. The PD-1/PD-L1 pathway may be involved in regulating T cell homeostasis and establishing feto-maternal tolerance [51]. PDL2 expression is increased in pregnant patients together with IL10. Pregnant MS patients increase *IL10*, *CD274* (*PDL1*) gene expression, and *PDCD1LG2* (*PDL2*) mRNA during the third trimester, resulting in a strong antiproliferative effect on CD4 T cells. This explains some of the amelioration in relapse activity during the third trimester of pregnancy [22]. *PDCD1LG2* genes are related to the cell adhesion molecule pathway. This pathway is crucial in T cell trafficking, activation, and function [52].

*CD274* and *PDCD1LG2* are related to T cell infiltration and the composition of the immunosuppressive microenvironment [53]. Cytokines are regulators produced by various types of cells. Cytokine–cytokine receptors, activating a signaling cascade, play a role in health and are crucial during immunological and inflammatory responses to diseases. However, they also combine with other cytokines in non-linear interactions [54]. The IL-10/IL-10 receptor axis plays a vital role in attenuating neuroinflammation in MS, and increased IL-10 has been associated with a positive response to disease-modifying therapy for MS. IL-10 acts directly on CD4+ T cells by inducing their energy, suppressing the expansion of pathogenic Th17 cells, and promoting the regulatory activity of CD4+ Foxp3+ regulatory T cells (Tregs) and type 1 CD4+ regulatory T cells (TR1) [55]. IL-10 plays an essential role in the formation and maintenance of immune tolerance in pregnancy [56,57].

The T cell receptor (TCR) requires mediators for its signaling. One of the critical mediators is the Vav family. *VAV3* participates in TCR signaling by promoting serum response element-mediated gene transcription. *VAV3* requires CD28 co-stimulation. Furthermore, Vav proteins transcriptionally regulate IL-2 promoter activity by acting on several transcription factors, including the nuclear factor of activated T cells (NFAT), NF-B, activator protein-1, and serum response factor [58]. Furthermore, *Vav3* is of great importance during pregnancy. The *VAV3* gene is activated in forebrain progenitors and can alter cell adhesion and division [59]. IL-8 is involved in the implantation process. IL-8 has been detected in placenta, amnion, the cervix, and the choriodecidua. IL-8 participates in recognizing placenta–fetus alloantigen by immune cells, and they can act as growth factors for the fetus [60].

## 7. Conclusions

An in-depth comprehension of the intricate interplay between genetic and molecular factors during pregnancy and the pathogenesis of MS could yield valuable insights for developing targeted therapeutic interventions or management strategies.

## Figures and Tables

**Figure 1 ijms-25-06741-f001:**
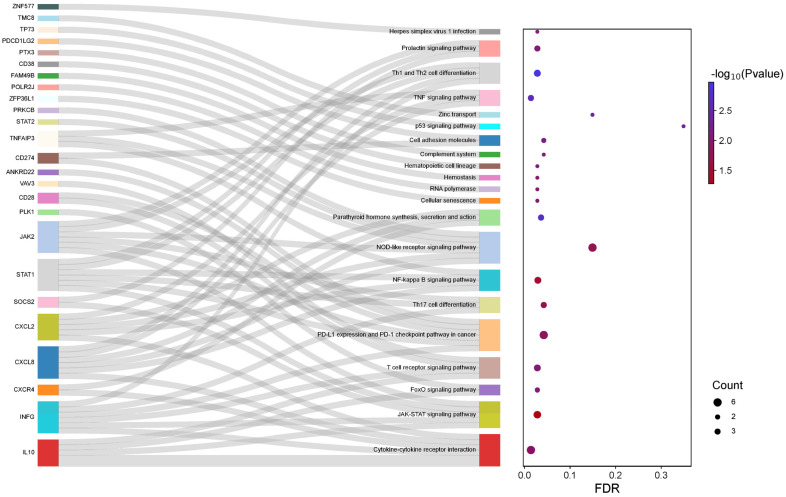
Using KEGG data, a Sankey Plot depicts the association between MS-related genes and signaling pathways. On the right side of the figure, there are 25 genes and 21 signaling pathways, with gray lines representing their relationships. The dot plot on the left side of the image displays the false discovery rate (FDR) on the *X*-axis. The color of the points is indicative of the *p*-value, normalized with −log^10^, and the size of the circles corresponds to the number of genes present in the signaling pathway (Count).

**Figure 2 ijms-25-06741-f002:**
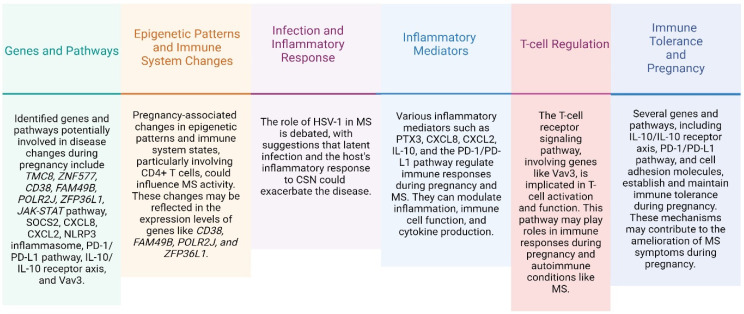
Bioinformatics research regarding potential genetic and molecular mechanisms. This figure provides insights into the underlying biological pathways that may contribute to the observed outcomes. Created with Biorender.com.

**Figure 3 ijms-25-06741-f003:**
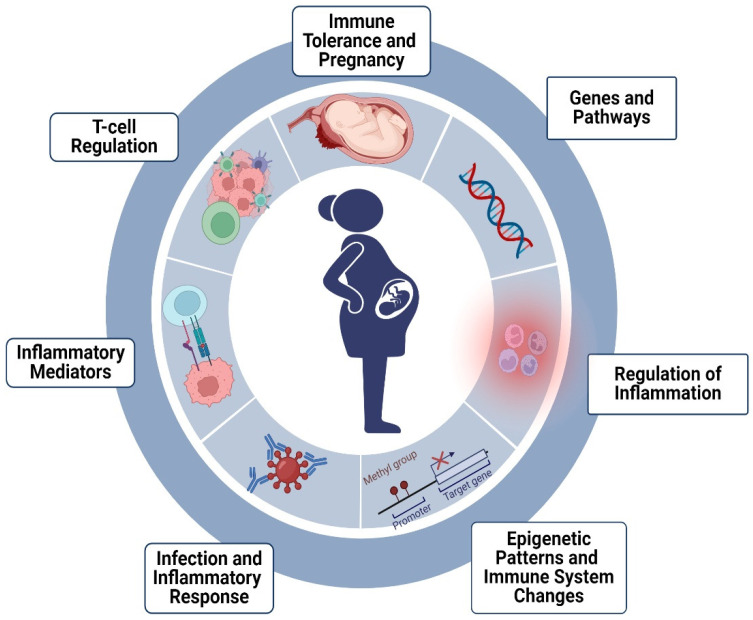
Genetic and molecular factors during pregnancy and in MS pathogenesis. Below is a breakdown of the important genetic and molecular factors that play a role during pregnancy and in the pathogenesis of MS. Created with Biorender.com.

**Table 1 ijms-25-06741-t001:** MS disability and relapses during pregnancy. Summary of studies conducted during pregnancy in women with MS.

Author	Year	Type	Sample Size	Disability and Relapses
[8]	2006	Original	254 women with multiple sclerosis.	Decrease in the relapse/year rate during the third trimester of pregnancy (0.2) versus a year before pregnancy (0.7) and an increase in the first three months after delivery (1.2). Disability progression steadily increased mean disability levels during the entire study period.
[9]	2011	Literature review and meta- analysis	1221 pregnancies reported in 13 papers.	During the year preceding pregnancy, women presented 0.435 relapses/year. During pregnancy, the relapse rate decreased to 0.26 relapses/year. After delivery, the relapse rate increased to 0.758 relapses/year.
[10]	2011	Original	302 pregnancies in 298 women.	The relapse rate significantly decreased during pregnancy, particularly in the third trimester, and increased in the postpartum, particularly in the first trimester (6.162, *p* = 0.001). The breastfeeding group (BG) had fewer relapses during pregnancy (0.06 vs. 0.14; *p* = 0.041) and 12 months after delivery (0.35 vs. 0.66; *p* = 0.001). Patients in the BG group had significantly lower EDSS scores at conception (1.3 vs. 1.6; *p* = 0.004) compared with the no BG group.
[11]	2018	Retrospective database research	2158 eligible patients.	Odds of relapse (OR) declined significantly (*p* < 0.001) during pregnancy (OR 0.623), increased during puerperium (OR 1.710), and remained higher than pre-pregnancy levels for 6 months postpartum (OR 1.216). The odds of relapse requiring hospitalization increased significantly during the third trimester (OR 1.849; *p* = 0.0011) and the puerperium.
[12]	2020	Meta-analysis	6430 women were included.	The annual relapse rate fell from 0.57 pre-pregnancy to 0.36 in the first trimester, 0.29 in the second trimester, and 0.16 during thirst trimesters, with a postpartum rebound (0.85). Relapses/year increased in the 3-month puerperium relative to the pre-pregnancy year period; the increase was 0.22 relapses/year *p* < 0.001. The range of the median EDSS was 0.7–2.4.
[13]	2021	Systematic review and meta-analysis	1469 abstracts were screened; 1293 articles.	The relapses/year rate increased in the 3-month puerperium relative to the pre-pregnancy year period; the mean increase was 0.22 (*p* < 0.001). Disability does not change significantly after pregnancy.

**Table 2 ijms-25-06741-t002:** Expression of genes related to MS in pregnancy. Compilation of studies conducted on the expression of genes related to MS during pregnancy.

Author	Patients	Gene	Result
[18]	6 MS patients during pregnancy.	*IL-10/IFN-g*.	The *IL-10/IFN-g* ratio was increased during pregnancy, especially in the third trimester, and decreased during the postpartum period compared with the third trimester (*p* = 0.043).
[20]	7 MS patients and 5 healthy controls followed during the 1st, 2nd, and 3rd trimesters and after pregnancy.	*SOCS2*, *TNFAIP3*, *NR4A2*, *CXCR4*, *ZFP36L1*, *POLR2J*, *FAM49B*, and *STAG3L1*.	Altered expression was observed in 347 transcripts in non-pregnant MS patients compared to healthy non-pregnant controls. Expression changes that occurred during pregnancy reversed the previous imbalance, particularly for seven inflammation-related transcripts (*SOCS2*, *TNFAIP3*, *NR4A2*, *CXCR4*, *POLR2J*, *FAM49B*, and *STAG3L1*). The deregulation of gene expression returned to “normal” within the third month of gestation.
[21]	10 women with MS and 9 healthy women.	*CD64*, *IL8*, *CXCL2*, *CD38*, *PTX3*, *JAK2*, and *STAT1*.	Increased CD64 expression during pregnancy is indicative of enhanced innate immune functions. A significant increase was observed in MS patients directly after delivery in the expression of *JAK2* and STAT1 (*p* < 0.05) compared to before pregnancy or until the third trimester. No statistically significant differences were found in the expression of the other genes when comparing the initial sample with the third trimester sample in patients with MS.
[24]	47 women with MS.	574 genes associated with parity.	The majority of differentially expressed genes (85%) were upregulated among women with MS who had fathered children. A total of 16 of the 574 genes had been reported to be clinically actionable genes. The results were not conclusive due to the sample size.
[22]	5 pregnant RRMS patients, 5 healthy controls, 12 MS third trimester patients, 12 MS postpartum patients, and 12 MS patient with no treatment.	754 miRNAs.	21 miRNAs were differentially expressed in the 3rd trimester compared to the postpartum period in MS patients. miR-1 in samples from the 3rd trimester of pregnancy in the non-pregnant patients showed a suggestive downregulation of miR-1 (*p* = 0.023). Only miR-18a was differentially expressed and was 1.3-fold higher in the 3rd trimester than in the postpartum period of pregnancy.
[23]	12 non-pregnant healthy women, 23 pregnant women, 11 first trimester women, and 12 second trimester women.	22 genes.	Twenty genes were hypermethylated (e.g., *CD28*, *CD86*, *PRKCB*, *TP73*, *PLK1*, and *VAV3*) and two were hypomethylated (*SPTBN4*, *MAML3*) when comparing second trimester pregnant and healthy non-pregnant women. These changes could be an important immune regulatory mechanism during pregnancy.
[19]	11 pregnant women with MS during the 1st, 2nd, and 3rd trimesters and after pregnancy.	CD4+ (n = 13,440 genes); CD8+ (n = 13,448 genes) and CpGs (N = 740,552 for both cell types).	DNA methylation and RNA sequencing in CD4^+^ and CD8^+^ T cells revealed a prominent regulation, mostly peaking in the third semester and reversing postpartum, thus minoring the clinical course with improvement followed by a worsening in disease activity.
[25]	192 women with relapse-onset MS (nulligravida = 96, parous = 96).	2965 differentially methylated positions in the whole blood.	It found 22 differentially methylated positions and 366 differentially methylated genes in epigenetic changes associated with parity.

## Data Availability

The presented data are available in this article.

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
