# Peer review of "Pregnant Women with Multiple Sclerosis: An Overview of Gene Expression and Molecular Interaction Using Bioinformatics Analysis"

_ijms, 2024, doi:10.3390/ijms25126741_

Round 1

Reviewer 1 Report

Comments and Suggestions for Authors

General comments: 

Marquez-Pedroza et el. propose to review gene expression in MS, and to identify gene-related signalling pathways during pregnancy in MS. Despite the topic is of potential interest, there are several aspects that in my view prevent the publication of the manuscript:

 - it is unclear whether the aim of the the manuscript is to propose a review of the existing literature on the topic of gene expression, pregnancy and MS, or an original research paper. In both cases the methods section is poor. It should be clearly stated how the review was performed, which search criteria were used, and which selection criteria for the identified research papers were applied. If the manuscript is an original research paper, it should be clearly stated where do the stated 25 genes come from, how was the expression analyse (if we are talking about expression, which for me is at the moment unclear); in which tissue/cell type were the genes analysed? What is the purpose of the described analysis? Expressions such as “various aspects” should be avoided. Are these genes mutated? is their expression increased or reduced? in comparison to what?

- the introduction is too long for a original research article, and too superficial for a review article (see examples of imprecisions below).

- result section:  the question that the analysis is trying to answer should be clearly stated. Currently the results show only the visualisation of the signalling pathways of 25 genes, whose way of identification is not specified. In figure 2, the attribution to each of the mentioned molecular mechanisms is arbitrary. 

- the discussion at the moment is a mixed summary of each described gene and potential significance. HSV-1 is mentioned in row 206 and anywhere else; please provide explanation. 

Specific comments. 

Row 51: do the author refer to the Journal NEJM? If so, please refer to the specific paper

Paragraph 3 should be expanded; it is not clear if the impact of pregnancy on disability refers to the short (during pregnancy, short after) or long term. Please clarify and if necessary adjust

Row 57: what does the expression “decrease influence on their EDSS” mean?

In general, the paragraphs 1 to 5 could be fused; if the authors are not willing to do that, each of the paragraphs needs to be expanded to provide more detailed information (studies, data and so on) on each single topic

Row 65 (table title): the expression “studies of disability” is not English

Table 1 should be summarised: at the moment, columns “relapses” and “disability” are difficult to read, and the interpretation difficult to extrapolate

Row 68: it is not clear to me how and why the “regulation” of MS should be considered a Th1 cell-mediated disease. Please revise the English

Row 69: increase of Th2 relative to Th1 cells, where? How?

Raw 70: changes in chemokine receptors, where? 

Row 76: CXCR3 expression decreased by T CD4 and T CD8: are CD4 and CD8 that induce the decreased expression of CXCR3 or they are those cells whose expression is reduced? Also, is it protein or RNA expression?

Row 79: expression of IL10 in which cell type?

These are just some of the various imprecisions that should be improved before publication. 

Comments on the Quality of English Language

- English should be revised by a native speaker, as often times the wrong use of grammar leads to misinterpretation of the sentence.  

Reviewer 2 Report

Comments and Suggestions for Authors

Your Review is a good contribution to the field and your effort and dedication to producer this work is appreciated. There are some issues requiring revision:

(1) Please explain, what does NEJM mean (line 51). 

(2). In the section "Treatment during pregnancy", lines 63, 64, you cite a reference (number 11) from an article published in 2019, although the preceding reference dated 2018 (number 10) addresses this particular topic more specifically. The information provided is outdated. More recent studies Gkinos, Pharmaceuticals 2023, and Krysko, Curr Ther Options Neurol 2021. provide a somewhat different view and more actualized understanding of this issue.  

(3) In page 4, paragraphs 5-8, lines 99-128, require redesigning and making these segments clearer since the text mixes information on DNA methylation as the authors attempt to discuss this phenomenon as a multi-exertional epigenetic mechanism affecting diverse elements including hormones, chemo receptors, cytokines, and immune cells. Lines 110-117 do not mention methylation being responsible for the changes, but "overall increase of CD56+ NK cells and a decrease of CD56-NK cells" (?). 

(4) Apparently your analysis did not include (did not detect) gene interaction with virus more strongly associated to the MS mechanism, such as EBV, and VZV, rather than Herpes simplex-1 as your study revealed (line 161), I believe this deserves a commentary (lines 206-208).   

Comments on the Quality of English Language

Considering that your paper, if accepted, would be published in a journal with global outreach, a better utilization of the linguistic format is desirable. These are some suggestions respectfully submitted from this reviewer:

Lines 59 and 60: "This evidence (use situation or occurrence or phenomenon instead) can result in a brief improvement in EDSS (10)" omit 'and impairment'. The way this line reads in the text is confusing. 

Table 2: Change the pronoun "they" in several segments of the table. It would be more appropriate to use a neutral, more generic form. Example, instead of "They observed altered expression of ....", perhaps use instead "Altered expression was observed....etc.

Round 2

Reviewer 2 Report

Comments and Suggestions for Authors

None. Do not expect further discussion with the authors.

Comments on the Quality of English Language

 N/A